# Parameters Identification of the Anand Material Model for 3D Printed Structures

**DOI:** 10.3390/ma14030587

**Published:** 2021-01-27

**Authors:** Martin Fusek, Zbyněk Paška, Jaroslav Rojíček, František Fojtík

**Affiliations:** Department of Applied Mechanics, Faculty of Mechanical Engineering, VŠB—Technical University of Ostrava, 708 00 Ostrava, Czech Republic; martin.fusek@vsb.cz (M.F.); jaroslav.rojicek@vsb.cz (J.R.); frantisek.fojtik@vsb.cz (F.F.)

**Keywords:** Anand material model, material parameters, ABS-M30, indentation test, genetic algorithm

## Abstract

Currently, there is an increasing use of machine parts manufactured using 3D printing technology. For the numerical prediction of the behavior of such printed parts, it is necessary to choose a suitable material model and the corresponding material parameters. This paper focuses on the determination of material parameters of the Anand material model for acrylonitrile butadiene styrene (ABS-M30) material. Material parameters were determined using the genetic algorithm (GA) method using finite element method (FEM) calculations. The FEM simulations were subsequently adjusted to experimental tests carried out to achieve the possible best agreement. Several experimental tensile and indentation tests were performed. The tests were set up in such a way that the relaxation and creep behaviors were at least partially captured. Experimental tests were performed at temperatures of 23 °C, 44 °C, 60 °C, and 80 °C. The results obtained suggest that the Anand material model can also be used for ABS-M30 plastic material, but only if the goal is not to detect anisotropic behavior. Future work will focus on the search for a suitable material model that would be able to capture the anisotropic behavior of printed plastic materials.

## 1. Introduction

The presented article is based on a conference paper [1]. The previous paper is extended with new experimental data and a new interpretation of the procedure used for the identification of material parameters.

The research project at VSB—Technical University of Ostrava is concerned, among other things, with the design of suitable robot arms and manipulators manufactured from plastic materials using 3D printing, one of the most advanced methods of component manufacture [2]. The individual arms (in general components made of printed plastic) are exposed to different types of mechanical loading.

The design of a structure is often determined by results of topological optimization analysis [3]. The structure obtained from the topological optimization process can also be produced using 3D printing [2]. A topological optimization methodology called the solid isotropic material with penalization (SIMP) method and the level set method [4] are often available in commercial software. The evolutionary structural optimization (ESO) method can be counted among the methods that are independent of the material model [5], which iteratively removes or adds a finite amount of material.

In order to predict the behavior of these printed components, knowledge of the appropriate material model and its parameters is necessary (in addition to the load) for FEM analysis. The parameters of printed 3D structures are highly dependent on the 3D printing technology, the laying of the filament, and the setting of the printing process (e.g., temperature). This article describes a procedure for the determination of the material parameters of a printed structure manufactured using 3D printing with a Fortus 450mc (Stratasys Ltd., Eden Prairie, MN, USA) 3D printer [6]. The material used for printing is ABS-M30 [7].

Several material models for acrylonitrile butadiene styrene (ABS) are described in the literature. For the large strain deformation and fracture behavior, there are three material models with different levels of complexity: (1) the Drucker–Prager yield function; (2) the Raghava yield function; (3) the Gurson yield function. These three models are compared in [8]. In [9], the isotropic Drucker–Prager yield criterion was used for the behavioral simulation of three semicrystalline polymers: high-density polyethylene (HDPE), polypropylene (PP), and polyamide 6 (PA 6). The Johnson–Cook model was used in [10] for the impact test. A constitutive model for polymers can also be found in [11]. The Anand model [12] was developed for aluminium alloys with viscoplastic behavior and is available in most FEM programs, which makes it advantageous for practical engineering uses. Regarding the thermoplastics used in 3D printing technologies, several material models were tested, and the Anand model gave surprisingly good results. Although the Anand model was developed to describe the viscoplastic behavior of metallic materials, its formulation is not limited to one material group only. It does not use the classical description used in other models, but rather it works with deformation resistance as an internal variable.

Based on the previous study [1], the Anand material model [12,13] was selected for further use. The FEM solution was implemented in ANSYS© software (Ansys, Canonsburg, PA, USA) [14], which has an extensive library of material models. ANSYS© was used as the FEM solver and was executed via a module written in Python programming language [15] by the authors. This module is independent of the FEM solver and can also be used in other commercial software (e.g., MSC.Marc). MATLAB (The MathWorks, Inc., Natick, MA, USA) can also be used for the same purpose (see [16]). The so-called finite element model updating (FEMU) [17] method was used to determine the material parameters. Another possibility in material analysis is the use of a neural network (e.g., [18]). FEM results can then be used to train the neural network (see [16]).

In [19], results obtained using the inverse approach were compared for two methods: using the gradient-based method and the evolutionary algorithm for a thermoelastic–viscoplastic material model. The article used the gradient method, which combines the steepest descent gradient and the Levenberg–Marquardt algorithm. The evolutionary algorithm for the real search space was described in the article mentioned. Both methods were able to determine the material parameters. The evolutionary method involves smaller relative error levels compared to the gradient-based method. On the contrary, the gradient method involves lower calculation complexity. The group of evolutionary algorithms also includes the genetic algorithm (GA), which is used in modified form in this work.

The influence of a certain parameter on a result is measured by the sensitivity. The sensitivities are described in [20] as “the partial derivatives of the output functions with respect to the parameters”. The modification of the sensitivity calculation was tested in the article mentioned and the resulting values were used to select parameters suitable for the identification. The time required for the process used for identification of the material parameters depends on the number of parameters identified and the number of experiments simulated. A low value for the sensitivity of a parameter may also indicate inappropriate or insufficient selection of experiments for the particular material model.

The proposed procedure is aimed at the identification of parameters in material models with larger numbers of material parameters and experiments. A procedure based on the selection of appropriate parameters for the identification process has not been published in the literature yet.

The main idea of the article is to demonstrate a modern approach for the determination of material parameters (not only 3D-printed materials) based on the use of a proper combination of standard laboratory tests at elevated temperatures and modern numerical optimization methods using custom and commercial software.

## 2. Experiments

Tensile tests, graded tensile tests, and indentation tests were performed. The experiments were performed on a Testometric M500-50CT (LABOR machine, s.r.o., Otice, Czech Republic) testing machine at four different temperatures (23 °C, 44 °C, 60 °C, and 80 °C). The testing machine used was equipped with a furnace (see Figure 1a). The machine was equipped with a strain gauge force sensor with a measuring range of 50 kN and a measuring accuracy of ±10 N. The maximum tensile force of the machine was 50 kN. The samples were clamped with pneumatic clamping jaws. The magnitude of the clamping force was set by the pressure in the system.

The specimen shown in Figure 2 was used for the tensile tests, graded tensile tests, and indentation tests. Figure 1b schematically shows the position of all specimens in the printing chamber during 3D printing on a Fortus 450mc printer and the method used to lay individual layers of ABS-M30 material.

After printing, the samples were stored in a dry and dark place for one month and then the necessary experiments were performed.

### 2.1. Tensile Tests

All tensile tests were deformation-controlled. Simple tensile tests were performed at four different temperatures (23 °C, 44 °C, 60 °C, and 80 °C) and at three different rates of deformation (0.017 mm s^−1^, 0.167 mm s^−1^, and 1.667 mm s^−1^) until specimen failure. Figure 3a shows the force against the displacement at 44 °C, 60 °C, and 80 °C, at a constant deformation rate of 0.017 mm s^−1^ (the constant test parameters are shown in the upper right corner of the diagram). It can be seen from the figure that the maximum force value decreased with increasing temperature, however at the beginning of the test the courses of the force were quite similar. One experiment was performed at room temperature, but with the highest deformation rate of 1.667 mm s^−1^ (see Figure 3b). Figure 3c,d shows the force dependence at 44 °C and 60 °C and at the specified deformation rates. The magnitude of the rate of the deformation had no significant influence on the force course at the beginning of the tests, but led to higher maximum force values at the end of the tensile tests.

### 2.2. Graded Tensile Tests

Graded tensile tests were carried out at three different temperatures (44 °C, 60 °C, and 80 °C), with the same deformation step size of 0.25 mm (see Figure 3e). The specimen was elongated by 0.25 mm at each step, at a deformation rate of 0.017 mm s^−1^. The time delay at the given strain value was always 60 s. This was done until the specimen failed. During the graded tensile test, the creep phenomenon was partially detected. Up to the value of the tensile force of approximately 800 N, there was no significant relaxation of the tension.

### 2.3. Indentation Tests

Indentation tests were carried out at 23 °C, 44 °C, 60 °C, and 80 °C (see Figure 3f) on the samples depicted in Figure 2. The area where the indentation test was carried out is shown by a dashed rectangle. The indenter was a steel sphere with a diameter of 5 mm. The indentation test consisted of three phases. First, the indenter was pressed into the material at a speed of 0.017 mm s^−1^ to a depth of 0.5 mm, then the time delay of 300 s followed, and finally the indenter returned to the starting position (relief) at the same speed.

## 3. Material Model

One of the material models used for viscoplastic materials is the Anand material model [12]. The Anand model was proposed for use in the analysis of the rate-dependent deformation of metals at high temperatures. The Anand viscoplastic model is pre-built in the commercial finite element software ANSYS©, and therefore it is much easier to use. The Anand model is typically used for solder alloys [21,22], however the previously mentioned benefits led the authors to test it for materials used for 3D printing (ABS-M30). The material model involves 11 material parameters: the Poisson ratio (µ), Young’s modulus (E), initial value of deformation resistance ($s_{0}$), activation energy/universal gas constant (Q/R), pre-exponential factor (A), stress multiplier ($x_{i}$), strain rate sensitivity of stress (m), hardening/softening constant ($h_{0}$), coefficient for deformation resistance saturation value (S), strain rate sensitivity of saturation (deformation resistance) value (n), and strain rate sensitivity of hardening or softening (a).

The Anand viscoplastic model was originally developed for material forming applications [12,13]. It is also applicable to general viscosity problems, which include the influence of the strain rate and temperature. Materials at elevated temperatures are highly dependent on the influence of the temperature magnitude and history, strain rate, and strain hardening. The Anand model is a complex material model that introduced an internal variable S (deformation resistance), a variable that represents the resistance against the plastic behavior of the material. This is different from other material models, and the decomposition of the individual components of the deformation is not straightforward.

The rate of plastic deformation is described using the following relationship:(1)$${\dot{\varepsilon }}_{\mathrm{pl}}={{\dot{\varepsilon }}^{a}}_{\mathrm{pl}}\left(\frac{3}{2}\;\frac{S}{q}\right),\;$$
where ${\dot{\varepsilon }}_{\mathrm{pl}}$ is the tensor of the inelastic strain rate and ${{\dot{\varepsilon }}^{a}}_{\mathrm{pl}}$ is the rate of accumulated equivalent plastic strain. Here, ${{\dot{\varepsilon }}^{a}}_{\mathrm{pl}}$ is given by the equation:(2)$${{\dot{\varepsilon }}^{a}}_{\mathrm{pl}}={\left(\frac{2}{3}{\dot{\varepsilon }}_{\mathrm{pl}}:{\dot{\varepsilon }}_{\mathrm{pl}}\right)}^{\frac{1}{2}},$$
where the operator “:” stands for inner product of the tensors. S is the deviator of the Cauchy stress tensor, which can be expressed using the following relation:(3)S=σ−pI,
where σ is the Cauchy stress tensor and p is defined as one-third of the trace of the tensor matrix σ, as in the following relation:(4)$$p=\frac{1}{3}\mathrm{tr}\left(\sigma \right).$$

Here, I represents a second-order unit tensor. The quantity q is the equivalent stress according to the following relation:(5)$$q={\left(\frac{3}{2}S:S\right)}^{\frac{1}{2}}.$$

The rate of accumulated plastic deformation depends on q and on the internal state variable s. This dependence can be expressed by the following equation:(6)$${{\dot{\varepsilon }}^{a}}_{\mathrm{pl}}={\mathrm{Ae}}^{\left(-\frac{Q}{R\theta }\right)}{\left\{\sinh\xi \frac{q}{s}\right\}}^{\frac{1}{m}},$$
where A, ξ, and m are the model constants; Q is the activation energy; R is the universal gas constant; θ is the absolute temperature; and s is the internal state variable.

Equation (6) indicates that plastic deformation occurs at any stress level. This contrasts with other theories of plasticity that use areas of plasticity (yield function). Classical theories assume that plasticity occurs above a certain stress value, otherwise the deformations are elastic.

The development of the internal state parameter s is described as follows:(7)$$ṡ=⊕h_{o}{\left|1-\frac{s}{s^{*}}\right|}^{a}{{\dot{\varepsilon }}^{a}}_{\mathrm{pl}},$$
where a and $h_{o}$ are constants, while $s^{*}$ represents the saturated value of the internal parameter. The ⊕ operator is defined to return +1 if $s\leq s^{*}$, otherwise it will return −1. The effect of softening or hardening is included in the model by this operator. The saturation values of $s^{*}$ depend on the rate of equivalent plastic deformation ${{\dot{\varepsilon }}^{a}}_{\mathrm{pl}}$ and can be expressed as follows:(8)$$s^{*}=\hat{s}{\left\{\frac{{{\dot{\varepsilon }}^{a}}_{\mathrm{pl}}}{A}e^{\left(\frac{Q}{R\theta }\right)}\right\}}^{n},$$
where $\hat{s}$ and n represent constants.

Expression (7) shows that the development of the parameter s depends on the rate of the equivalent plastic deformation, and at the same time on the current state of the internal state parameter s.

## 4. Theoretical Description of the Identification Procedure

The material parameter identification can be described mathematically as:(9)$$f\left(X\right)=\mathrm{minimum},\;\mathrm{subject}\;\mathrm{to}\;g_{j}\left(X\right)>0,j=\left\{1,2,3,\ldots ,{\;N}_{C}\right\},$$
where $f\left(X\right)$ is an objective function, X represents a vector of material parameters, $g_{j}\left(X\right)$ is j-th constrain function, and $N_{C}$ is the number of constraint functions. For minimalization, the finite element model updating (FEMU) approach was used, which is described in [23]. Schematically, the parameters of the material model were determined by repeated optimization calculations with custom software written in Python programming language. The solution procedure was as follows: FE models of the experiments were created at first. The models were created in commercial software (ANSYS©) as boxes, where the inputs were values of the material parameters and the outputs were values corresponding to loaded or measured data (in this paper these were a force, a displacement, and a current time). The difference between the simulation outputs and the measured data defines the value of an objective function. The simulations were solved in a cycle, whereby the inputs were changed with respect to the minimized value of the objective function. All experiments were deformation-controlled, but forces were used for formulation of the objective function. The difference between experimental data and data from the simulation model for one experiment was solved as an individual objective function $f_{i}\left(X\right)$ for the i-th experiment:(10)$$f_{i}\left(X\right)=\frac{\sum_{j=1}^{N_{i}}\left|F_{j}^{\mathrm{EXP}}-F_{j}^{\mathrm{FEM}}\left(X\right)\right|}{\sum_{j=1}^{N_{i}}\left|F_{j}^{\mathrm{EXP}}\right|},\;i=\left\{1,2,3,\ldots ,\;N\right\},$$
where $N_{i}$ is the number of measurement points for the i-th experiment, $F_{j}^{\mathrm{EXP}}$ is an experimental force, $F_{j}^{\mathrm{FEM}}\left(X\right)$ is the force obtained from the simulation, and N is the number of experiments.

The value of objective function for all experiments ($f\left(X\right)$) was solved as:(11)$$f\left(X\right)=\sqrt{\frac{\sum_{i=1}^{N}f_{i}{\left(X\right)}^{2}}{N}},$$

A genetic algorithm (GA) was used to identify the material parameters, as in [1]. A GA for the identification of material parameters was used in [24]. The GA was used because it is not dependent on the material model. A detailed description of the GA can be found in [25]. A chromosome is defined by a vector of genes:(12)$$X=\{p_{i},\;i=\left\{1,2,3,\ldots ,{\;N}_{p}\right\}\},$$
where $p_{i}$ is the i-th gene, which corresponds to the i-th material parameter, and $N_{p}$ is the number of parameters.

An initial population was given by $N_{G}\;$ solutions, which was generated randomly by using the hill-climbing algorithm [26] from an initial chromosome. For the i-th gene we used:(13)$$p_{i}{}^{\mathrm{New}}=p_{i}{}^{\mathrm{Old}}\cdot (1+\;\mathrm{Rand}\left(-k_{\mathrm{HC}},{\;k}_{\mathrm{HC}}\right)),\;i=\left\{1,2,3,\ldots ,\;N\right\},$$
where $p_{i}{}^{\mathrm{Old}}$ is the original value of the i-th gene, $p_{i}{}^{\mathrm{New}}$ is the proposed value of the i-th gene, and the Rand function generates random values with uniform distribution from the interval given by $\pm k_{\mathrm{HC}}$.

The constraint functions are defined as:(14)$$g_{j}\left(X\right)=p_{j},j=\left\{1,2,3,\ldots ,{\;N}_{p}\right\},$$
where the number of constraint conditions $N_{c}$ is the same as the number of parameters $N_{p}$.

In each cycle a new individual is created, its objective function value is calculated, and then the individual is added to the population. Regarding the quality of an individual, the solution is determined by the value of the objective function (Equation (11)). The new individual is created in two ways:With a 40% probability using the hill-climbing algorithm for the best individual in the population (13);With a 60% probability using crossover.

The child chromosome was created from three chromosomes of parents, whereby the parents were selected randomly (uniform distribution) from the population. The parents were sorted by the value of the objective function and are indicated by a superscript of 1, 2, or 3; parent 1 had the best (the lowest) objective function and parent 3 had the worst (the greatest) objective function. The child value of the gen ($p_{i}{}^{\mathrm{Child}}$) is calculated as:(15)$$p_{i}{}^{\mathrm{Child}}=p_{i}{}^{\mathrm{Best}}+\Delta p_{i}.\;\;i=\left\{1,2,3,\ldots ,\;N\right\},$$
where $\Delta p_{i}$ is the gene modification value. The modification value is calculated from parent genes as follow:(16)$$\Delta p_{i}=k\left(p_{i}{}^{1}-p_{i}{}^{2}\right),\;\mathrm{or}\;\Delta p_{i}=k\left(p_{i}{}^{1}-p_{i}{}^{3}\right),\;\;i=\left\{1,2,3,\ldots ,\;N\right\},$$
where equations are selected randomly (with 70% and 30% probability); $p_{i}{}^{1}$, $p_{i}{}^{2}$, $p_{i}{}^{3}$ are the gen values of the first parent, second parent, and third parent, respectively; k is a growth coefficient $\left(1-2\right)$.

The size of the resulting gene values is controlled. If the gene values change too little $\Delta p_{i}<p_{i}{}^{\mathrm{Best}}\;\left(1\pm \;0.0001\right)$ or too much $\Delta p_{i}\;>p_{i}{}^{\mathrm{Best}}\;\left(1\pm \;0.05\right)$, then the new gene values are generated randomly using the hill-climbing algorithm with $k_{\mathrm{HC}}=0.001$.

Parameters suitable for identification are selected using sensitivity analysis. This sensitivity analysis shows an effect of the parameter on the value of the objective function with respect to the other parameters. The sensitivity analysis is performed from Np+1 calculations as follows:(17)$$f=f\left(X\right),\;f_{+\Delta i}=f\left(\left\{p_{1},p_{2},\ldots ,{\;p}_{i}\left(1+k_{\mathrm{Sen}}\right),\ldots ,p_{\mathrm{Np}}\right\}\right),$$
where $k_{\mathrm{Sen}}$ is a sensitivity coefficient $\left(0.05-0.001\right).$ The sensitivity value $S_{i}$ is calculated according to the equation:(18)$$S_{i}=\frac{100}{k_{\mathrm{Sen}}}\;\left|1-\frac{f_{+\Delta i}}{f}\right|\;\left[\;\%\;\right],\;i=\left\{1,2,3,\ldots ,\;\mathrm{Np}\right\}.$$

Parameters whose sensitivity values are greater than the critical value $S_{\mathrm{Krit}}$ are selected for identification:(19)$$S_{i}>S_{\mathrm{Krit}}\to {\mathrm{identify}\;p}_{i},\;i=\left\{1,2,3,\ldots ,\;\mathrm{Np}\right\}.$$

In later phases of identification, it is advisable to check if the given parameter $p_{i}$ is not in the local minimum of the objective function. Here, the number of required calculations is 2Np+1:(20)$$f_{+\Delta i}>f<f_{-\Delta i},\;i=\left\{1,2,3,\ldots ,\;\mathrm{Np}\right\}.$$

## 5. Description of the Identification Process

Ten experiments were used for identification (see Figure 3a–f), however the experiments at 44 °C were excluded from the identification process and were used only for validation. The assignment of the experiments to the values of the individual objective functions is shown in Table 1.

In Table 1, the temperatures are denoted as T_1_ = 23 °C, T_2_ = 44 °C, T_3_ = 60 °C, and T_4_ = 80 °C; the deformation rates are ${ẋ}_{1}\;=\;0.017\;{\mathrm{mm}\;s}^{-1}$, ${ẋ}_{2}\;=\;0.167\;{\mathrm{mm}\;s}^{-1}$, and ${ẋ}_{3}\;=\;1.667\;{\mathrm{mm}\;s}^{-1}$; ID denotes the indentation tests and GT denotes the graded tensile tests (see Chapter 2). Data from the tests at temperature T_2_ were used to validate the parameters.

The initial values of parameters are shown in Table 2.

The value of the modulus of elasticity E is temperature-dependent, which is shown in Figure 3a. For this reason, three values of the modulus of elasticity $E_{20}$, $E_{50}$, and ${\;E}_{80}$ (for 20, 50, and 80 °C) were added to the material model. The temperature chamber made it impossible to use optical methods (e.g., the digital image correlation method (DIC)) [27] to measure the Poisson number, so this parameter was excluded from identification.

Chromosome (12) consists of these material parameters, whereby the individual genes correspond to the individual material parameters:(21)$$X=\left\{E_{20},E_{50},E_{80},s_{0},\frac{Q}{R},A,x_{i},m,h_{0},\hat{S},n,a\right\}.$$

The hill-climbing algorithm was used with the parameter $k_{\mathrm{HC}}=0.001$, the genetic algorithm was used with the parameter $N_{G}=10$, and the growth coefficient k=1.5 was used in both cases. The sensitivity coefficient was $k_{\mathrm{Sen}}=0.01$. The sensitivity value shown in (18) and the criterion shown in (20) were recalculated. The way in which the model parameters were determined is described by the following three tables. Table 3 contains the parameter values during the solution.

The values for the objective function during the solution are shown in Table 4 and the sensitivity values are shown in Table 5. The initial parameters for identification (values of the objective function) are given in line 0 of Table 3. The sensitivity was calculated from “line 0” values, and the resulting values are shown on line 1 of Table 5. The genetic algorithm was run 200 times in every identification cycle. This process was repeated in lines 2 to 5. Line 5 in Table 4 and Table 5 shows the resulting parameter and error values. The fulfilment of criterion (20) is indicated in Table 5 as U. The resulting values (line 5, Table 4) were used to calculate the sensitivity (line 6* in Table 5). Values in line 6* were no longer used for further calculation. This shows that the minimum given by the coefficient $k_{\mathrm{Sen}}$ had not been reached yet. Table 5 shows in the last step (line 5) a decrease of the objective function value by 0.1% after 200 calculation cycles; the identification process was, therefore, completed.

## 6. Results

The obtained final parameter values of the Anand material model are summarized in Table 6 and the FEM analyses were performed with these material parameters. The value of the Poisson ratio was µ=0.33. Figure 4 and Figure 5 show a comparison of experiments with results from the FEM analysis.

Very good agreement can be seen between the experiments and the FEM solution. Different behavior just before the failure of the specimen is shown in Figure 5a,b,d–f,h. This behavior is less significant under lower temperatures and higher deformation rates. The difference can be seen in Figure 5a,f especially (temperatures of 60 and 80 °C and deformation rate of 0.017 m s^−1^). This effect was not captured in the simulations.

## 7. Validation

Validation testing was carried out on a series of experiments at 44 °C. Experiments at this temperature were not used for material parameter identification. The results are shown in Figure 6. In Figure 6, a very good agreement can be seen between FEM solutions and experiments. The greater error occurs just before the failure of the specimen, as also mentioned in the previous chapter. The conclusion is that the material models with those material parameters do not accurately describe the mechanical behavior shortly before the sample ruptures. In order to describe this area of difference better, more experiments will be necessary, especially focused on this small area. If no experiments of this kind are carried out, the influence of this small area on the value of the objective function is negligible.

## 8. Discussion

The GA modification was used to identify the parameter values. In order to reduce the number of calculation cycles and individuals in the GA population, the number of parameters was reduced by means of sensitivity control. The sensitivity calculation was performed at the beginning of the solution, and then again when the GA convergence rate decreases significantly.

Further sensitivity tests showed that if some parameters are changed there is still a decrease, while others are at their minima in the range tested (see table of resulting sensitivity U). In subsequent cycles, the value of the objective function gradually decreased in the range of 0.01% to 100 GA steps. This value decreased further in each cycle. The resulting effects of these calculations on the value of the objective function and the values of parameters were negligible, and therefore are not described in this contribution.

The Anand constitutive model used in this article contained 9 parameters and 3 moduli of elasticity, which were identified for temperatures of 20, 50, and 80 °C. Figure 3a shows the influence of temperatures on the value of E. This dependence appeared at all speeds tested, so it was included in the identification, although its effect was not significant. Input values of the material parameters were taken from [1] for the ABS-M30 material and determined at a temperature of 23 °C and displacement rate of 1 mm s^−1^ from tensile tests and indentation tests, while the Poisson’s ratio was determined from tensile tests with DIC. These experiments were not used for identification in this contribution because they showed large differences (originally E=1780 MPa, new value E=1100 MPa) in the behavior compared to the results presented above. The authors believe that this was due to a difference in the supply of the material and different 3D printing settings.

In [1], indentation and DIC measurement data were also used to identify and validate the results. When using a temperature chamber, it was not possible to perform DIC measurements—more extensive modifications of the experimental equipment would be necessary. In the case of DIC, this mainly involves determining the value of the Poisson ratio, which was, therefore, taken from [1]. Indentation tests describe the behavior of multi-axial stresses in the area above the yield point in the compression area. Tensile tests carried out at lower temperatures and higher speeds (Figure 3b,c) show almost linear behavior.

The use of different types of experiments (e.g., tensile tests, indentation tests) for identification or at least for validation is the key for the further use of the identified parameters. Indentation, tensile, and graded tensile tests at 44 °C were used for validation (see [28]). Components produced using 3D printing are not only loaded by tension, so the authors assume that after modifying the experimental equipment, a different set of experiments will be performed. On the other hand, conventional components are loaded in the area corresponding to the first half of the loading curve, where tensile tests in particular show very good agreement with the FEM solution.

## 9. Conclusions

The obtained results show the possibility of finding the material model parameters using indentation or tensile tests. The results of the FEM calculations are in good agreement with the data obtained experimentally. Attention will be paid to the modification of the test equipment in order to enable DIC measurements. The Anand material model can be used for specimens loaded at different temperatures. It was found that the Anand material model describes the behavior of the ABS-M30 material very well with respect to different temperatures and loading speeds. On the other hand, the Anand model is not able to capture the anisotropic behavior of the investigated material sufficiently. For tests on specimens printed in the most unfavorable position (related to the load capacity of the specimen), the material model parameters identified as described above will give conservative results and the model can be used to design components. Studies on a more advanced material model that includes the effects of material anisotropy and a damage model are currently in progress. The proposed identification methodology is not dependent on the material model.

## Figures and Tables

**Figure 1 materials-14-00587-f001:**
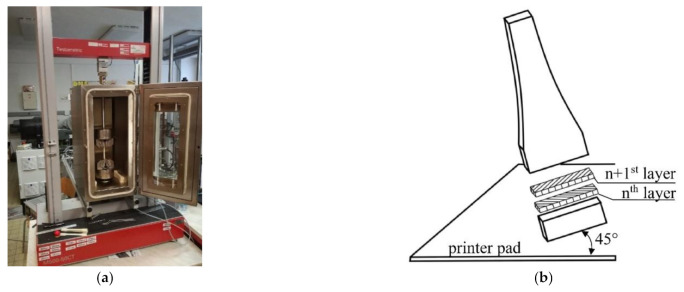
(**a**) The Testometric M500-50CT tensile testing machine equipped with a temperature chamber. (**b**) The position of specimens during 3D printing and the method used to layer the material.

**Figure 2 materials-14-00587-f002:**
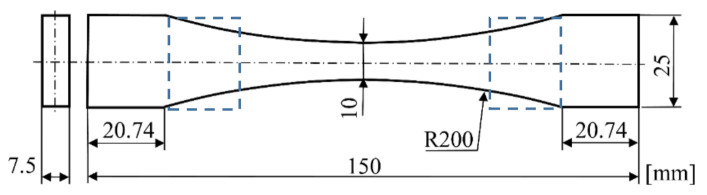
Specimen shape for simple and graded tensile tests.

**Figure 3 materials-14-00587-f003:**
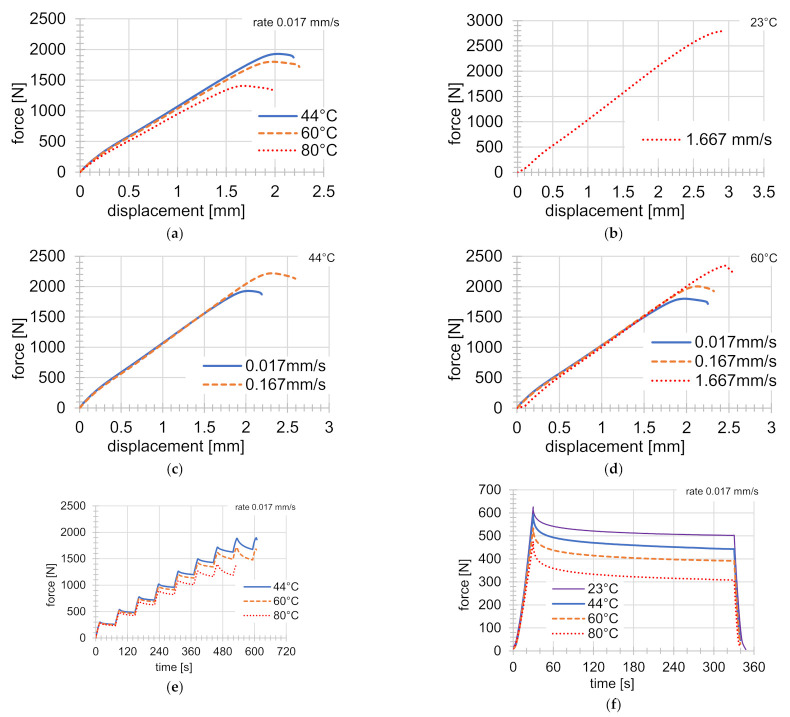
(**a**) Tensile tests at temperatures of 44 °C, 60 °C, and 80 °C, with constant rates of deformation at 0.017 mm s^−1^. (**b**) Tensile tests at 23 °C and at a constant rate of deformation of 1.667 mm s^−1^. (**c**) Tensile tests at a temperature of 44 °C, with two different rates of deformation. (**d**) Tensile tests at 60 °C, with three different rates of deformation. (**e**) Graduated tensile tests under three different temperatures of 44 °C, 60 °C, and 80 °C. (**f**) Indentation tests with time delay at temperatures of 23 °C, 44 °C, 60 °C, and 80 °C, with a constant indenter rate of 0.017 mm s^−1^.

**Figure 4 materials-14-00587-f004:**
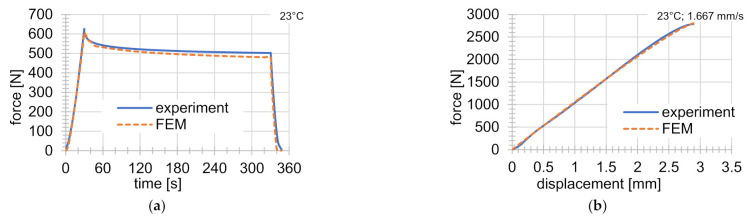
Comparison of experiments with FEM solutions: (**a**) indentation test at 23 °C; (**b**) tensile test at 23 °C and rate of deformation of 1.667 mm s^−1^.

**Figure 5 materials-14-00587-f005:**
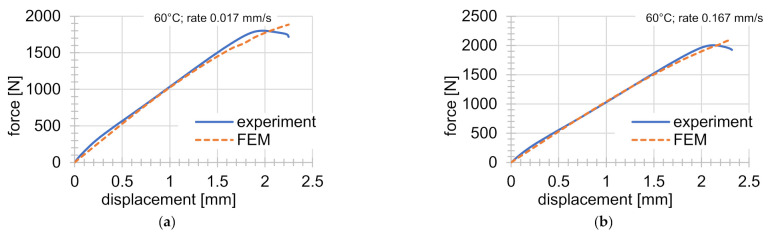
Comparison of experiments with FEM solutions (continuation): (**a**) tensile test at 60 °C and rate of deformation of 0.017 mm s^−1^; (**b**) tensile test 60 °C and rate of deformation of 0.167 mm s^−1^; (**c**) indentation test at 60 °C; (**d**) tensile test at 60 °C and rate of deformation of 1.667 mm s^−1^; (**e**) graded tensile test at 60 °C; (**f**) tensile test at 80 °C and rate of deformation of 0.017 mm s^−1^; (**g**) indentation test at 80 °C; (**h**) graded tensile test at 80 °C.

**Figure 6 materials-14-00587-f006:**
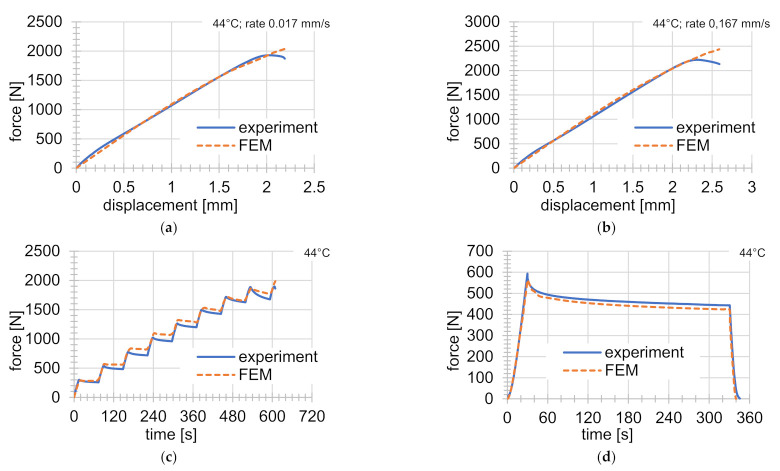
Comparison of FEM solutions with validation experiments at 44 °C: (**a**) simple tensile test at rate of deformation of 0.017 mm s^−1^; (**b**) simple tensile test at rate of deformation of 0.167 mm s^−1^; (**c**) graded tensile test; (**d**) indentation experiment.

**Table 1 materials-14-00587-t001:** Assignment of the individual objective functions to individual experiments.

$f_{1}\left(X\right)$	$f_{2}\left(X\right)$	$f_{3}\left(X\right)$	$f_{4}\left(X\right)$	$f_{5}\left(X\right)$	$f_{6}\left(X\right)$	$f_{7}\left(X\right)$	$f_{8}\left(X\right)$	$f_{9}\left(X\right)$	$f_{10}\left(X\right)$
T_1_, ID	T_1_$,\;{ẋ}_{3}$	T_3_$,{ẋ}_{1}$	T_3_$,{ẋ}_{2}$	T_3_, ID	T_3_$,\;{ẋ}_{3}$	T_3_, GT	T_4_$,\;{ẋ}_{1}$	T_4_, ID	T_4_, GT

**Table 2 materials-14-00587-t002:** Initial parameters of the Anand material model.

E $\left[MPa\right]$	μ $\left[-\right]$	$s_{0}$ $\left[MPa\right]$	Q/R $\left[K\right]$	A $\left[1/s\right]$	$x_{i}$	m $\left[-\right]$	$h_{0}$ $\left[MPa\right]$	$\hat{S}$ $\left[MPa\right]$	n $\left[-\right]$	a $\left[-\right]$
1780	0.33	19.54	8350	3134	5.18	0.2466	183,245	31.0	0.0098	1.524

**Table 3 materials-14-00587-t003:** Parameter values with three moduli of elasticity during identification.

#	$E_{20}$ $\left[MPa\right]$	$E_{50}$ $\left[MPa\right]$	$E_{80}$ $\left[MPa\right]$	μ $\left[-\right]$	$s_{0}$ $\left[MPa\right]$	Q/R $\left[K\right]$	A $\left[1/s\right]$	$x_{i}$	m $\left[-\right]$	$h_{0}$ $\left[MPa\right]$	$\hat{S}$ $\left[MPa\right]$	n $\left[-\right]$	a $\left[-\right]$
0	1780	1780	1780	0.33	19.5	8350	3134	5.18	0.247	183,245	31.0	0.0098	1.524
1	1175	1161	933	-	19.5	9486	3134	5.75	0.24	183,245	41.4	0.0098	1.897
2	1175	1161	933		19.5	9486	3134	5.82	0.241	169,305	43.6	0.012	2.936
3	1175	1203	960		19.5	9486	3134	5.82	0.224	169,305	43.6	0.0152	2.936
4	1196	1203	997		18.0	9486	3048	5.82	0.224	146,821	43.6	0.0186	3.527
5	1196	1224	997		18.0	9486	3263	8.82	0.213	138,175	43.6	0.0226	3.527

**Table 4 materials-14-00587-t004:** The objective function values during identification.

#	$f_{1}\left(X\right)$ $\left[\%\right]$	$f_{2}\left(X\right)$ $\left[\%\right]$	$f_{3}\left(X\right)$ $\left[\%\right]$	$f_{4}\left(X\right)$ $\left[\%\right]$	$f_{5}\left(X\right)$ $\left[\%\right]$	$f_{6}\left(X\right)$ $\left[\%\right]$	$f_{7}\left(X\right)$ $\left[\%\right]$	$f_{8}\left(X\right)$ $\left[\%\right]$	$f_{9}\left(X\right)$ $\left[\%\right]$	$f_{10}\left(X\right)$ $\left[\%\right]$	$f\left(X\right)$ $\left[\%\right]$
0	27	28	20	23	28	27	23	21	23	25	25
1	7.1	7.9	6.3	7.2	4.9	6.6	8.5	9.9	3.7	7.0	7.1
2	5.5	7.6	5.7	7.3	3.7	6.9	5.6	9.5	4.7	5.4	6.4
3	5.7	7.7	4.6	6.5	3.7	6.2	6.3	8.4	4.5	5.1	6.0
4	5.2	7.9	4.4	6.4	4.0	6.3	4.9	7.0	4.4	4.5	5.6
5	4.2	7.9	4.2	6.3	3.4	6.3	4.9	7.0	4.5	4.0	5.5

**Table 5 materials-14-00587-t005:** Sensitivity values during identification. Parameters that fulfil the criterion are shown in bold.

#	$S_{E_{20}}$	$S_{E_{50}}$	$S_{E_{80}}$	$S_{s0}$	$S_{QR}$	$S_{A}$	$S_{xi}$	$S_{m}$	$S_{h0}$	$S_{S}$	$S_{n}$	$S_{a}$	$S_{Krit}$
1	**13.4**	**34.3**	**42.0**	4.1	**115.0**	6.3	**42.1**	**30.8**	3.3	**47.2**	4.2	**12.7**	10
2	U	U	U	1.0	U	0.2	**33.9**	**15.2**	**7.3**	**16.6**	**8.3**	**23.4**	5
3	3.9	**97.2**	**102.4**	4.4	U	U	U	**8.4**	1.4	U	**9.2**	1.8	5
4	**3.8**	U	**5.2**	**8.5**	U	**4.5**	U	U	**7.0**	U	**7.7**	**30.5**	0.1
5	U	**0.3**	U	U	U	**0.9**	U	**8.6**	**1.6**	U	**16.7**	U	0.1
6*	30.0	U	0.1	7.5	U	3.2	U	U	3.0	U	4.2	6.8	-

**Table 6 materials-14-00587-t006:** Final parameter values.

$E_{20}$ $\left[MPa\right]$	$E_{50}$ $\left[MPa\right]$	$E_{80}$ $\left[MPa\right]$	$s_{0}$ $\left[MPa\right]$	Q/R $\left[K\right]$	A $\left[1/s\right]$	$x_{i}$	m $\left[-\right]$	$h_{0}$ $\left[MPa\right]$	$\hat{S}$ $\left[MPa\right]$	n $\left[-\right]$	a $\left[-\right]$
1196	1224	997	18.0	9486	3263	8.82	0.213	138,175	43.6	0.0226	3.527

## Data Availability

Data can be provided upon request from the correspondent author.

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
