# Peer review of "Parameters Identification of the Anand Material Model for 3D Printed Structures"

_materials, 2021, doi:10.3390/ma14030587_

Round 1
Reviewer 1 Report
The paper deals with modeling of AM material, which is an interest for the industry. The work is well performed engineering work.
However, on the other hand, the work lacks novelty because of its strict engineering approach. The model fitting to experimental results is rather basic work in general, although here the authors provide additional usage of indentations etc.
At it is present format, the reviewer cannot promote its publication due to lack of novelty in the paper. A clear statement of novelty should be given with respect to the cutting edge recent advancements in the field. Furthermore, the paper would gain significant new prospects if the authors would also use a damage model to study the behavior of the material. This is important since the AM materials usually have certain amount of defects that can be considered to early phases of damage, hence the damage evolution with respect strength&life time is the essence. The reviewer suggest that the authors consider this possibility.
Reviewer 2 Report
The authors focused on the evaluation of material parameters of the Anand material model for ABS-M30 material, using Genetic Algorithm equipped with the FE numerical calculations. However, it needs further improvements to be suitable for a journal publication.
Reviewer 3 Report
The authors investigate the material behaviour of additively manufactured ABS-M30 samples. Experiments are conducted over a wide range of temperatures and a genetic algorithm is used to identify the parameters that provide the best fit to Anand’s creep model. The authors conclude that the fitting exercise enables reaching a good agreement with experiments but raise the point that material anisotropy is not captured. The paper is interesting and well-suited for Materials. However, this reviewer has the following concerns:
1) It is interesting (surprising?) that a constitutive model that is used for solder alloys is employed to capture the behaviour of a thermoplastic. The physics of creep in solder alloys and the mechanisms underlying the experiments are rather different. Why was this material model chosen in the first place?
2) It is not clear what the aim of the work is, in the way that it is formulated now. One can anticipate without any experiments/calculations that Anand’s model will not capture material anisotropy, as it assumes an isotropic solid. So the main conclusion is not very useful. Maybe what the authors should do is to re-frame the conclusions such that the main finding is that (if properly fitted) Anand’s model is able to capture well the experimental trends (except for material anisotropy of course).
3) Among all the ML tools available to fit experiments, the reviewer wonders why the use of GA and misses a comment relative to other approaches (e.g., Neural Networks, as pioneeringly used in Section 4.2 of this paper: Advances in Engineering Software, 105: 9-16 (2017))
Reviewer 4 Report
In this manuscript, the authors used Genetic Algorithm to estimate the parameters for the Anand material model developed in ANSYS. In general, the reviewer agrees with the methodology proposed in this manuscript. However, the verification step is insufficient because the authors compared the estimated model to the set of data that was used to estimate that model. Therefore, a new set of data is required to verify the model. Next, the authors should publish all experimental data that was used to estimate the model, so, the reader can reference it for future researches. Additionally, the introduction needs to be improved by giving the context in using the estimation method in developing the material model. Finally, this manuscript is recommended to resubmit for a major revision.
Round 2
Reviewer 2 Report
From the subject point of view the article can be accepted and the article can be accepted for publication in its current state.
Reviewer 3 Report
The authors have addressed satisfactorily all my comments. The work brings some interesting findings and deserves publication in Materials.